# Treatment completion of drug-resistant tuberculosis in Ethiopia: A perspective from healthcare users

**Ahmed Reshid Tusho**[1]*, **Sheila Theresa Mokoboto-Zwane**[2]

**1** Department of Tuberculosis and Leprosy, Oromia Health Bureau, Addis Ababa, Ethiopia, **2** Department of Health Studies, College of Human Sciences, University of South Africa, Pretoria, South Africa

* aciroo@yahoo.com, 62121901@mylife.unisa.ac.za

## Abstract

Drug-resistant tuberculosis remains a persistent public health threat. Maximizing successful treatment completion is a global health priority and a focus of the End TB strategy. Despite the implementation of programmatic management for drug-resistant tuberculosis in Ethiopia, there is limited understanding of the barriers related to successful treatment completion among Ethiopian patients. A qualitative study that is explorative, descriptive and contextual in nature was conducted to explore and describe the views and lived experiences of previously treated drug-resistant tuberculosis patients to gain an in-depth understanding of barriers to the successful completion of drug-resistant tuberculosis treatment. Six focus group discussion sessions with 42 purposively selected drug-resistant tuberculosis patients with previous treatment history and on retreatment regimens were conducted. The seven prominent themes revealed were: drug-related challenges encompassing adverse events and the lengthy treatment duration; clinical hurdles such as delayed consultation following prolonged illness, diagnostic delays, and suboptimal dosages; psycho-emotional difficulties including emotional trauma and facing stigma from both the community and healthcare professionals; socio-economic constraints; service-related issues such as interruptions in monitoring tests, inadequate follow-up, and accessibility barriers; patient-related factors such as a false sense of recovery and reverting to previously quit habits; and provider-related issues such as lack of responsiveness and ineffective communication. Addressing these factors demands policy-level decisions and programmatic guidance at the national level based on best practices, as well as good programmatic implementation from actors through regional and health facility-level structures.

## 1. Introduction

Drug-resistant tuberculosis (DR-TB) continues to be a public health threat. Resistance to rifampicin, the most effective first-line drug, is of greatest concern. The emergence of multidrug-resistant tuberculosis (MDR-TB) has made tuberculosis management more complicated than before. MDR-TB is a variant of TB caused by mycobacterium tuberculosis strain

**Data availability statement:** All relevant data are within the paper and Supporting information files.

**Funding:** The authors received no specific funding for this work.

**Competing interests:** The authors have declared that no competing interests exist.

resistant to Rifampicin and Isoniazid [1,2]. Both MDR-TB and rifampicin-resistant tuberculosis (RR-TB) require treatment with second-line drugs. Drugs used to treat DR-TB are more expensive, more toxic, and less effective than those used to routinely treat TB, making the treatment of DR-TB complex in a number of ways [3–5].

Patients on DR-TB treatment regimens need to be monitored for treatment response or failure and for their safety, using reasonable schedules of relevant clinical and laboratory testing. Response to treatment and toxicity is monitored through regular history taking, physical examination, chest radiography, special tests such as audiometry, visual acuity tests, electrocardiography and laboratory monitoring [6].

According to the Global TB Report 2022, in Ethiopia, among notified TB cases case adult women were 39%, adult men were 51%, and children were 10%. In the same report, the latest treatment outcome data show treatment success rates of 86% for the drug susceptible TB patient cohort and 68% for the MDR/RR-TB patient cohort [1].

Adherence to TB treatment has been recognised as an essential element in TB control programmes, and poor adherence is an extreme risk for their success [7]. Non-adherence to therapy can cause relapse, continued transmission and the increase of extensive drug resistance [7]. Previous studies indicate that a number of factors may contribute to patient adherence [7,8].

Further, when it comes to DR-TB, the anecdotal evidence shows that treatment interruption and non-adherence to treatment make patients more susceptible to TB, resulting in unsuccessful treatment outcomes. There is, therefore, a dire need for further exploration of the variables related to treatment completion. The world health organization (WHO) indicates that treatment interruption is one of the factors contributing to decreased treatment success [6].

Adherence is a major problem in the treatment of DR-TB because of the long duration of treatment and adverse effects of second line drugs [9]. Non-adherence to TB treatment has serious negative consequences, resulting in high defaulter rate, further resistance and treatment failure [10].

Maximizing successful treatment completion in DR-TB is a global health priority and focus area of the End TB strategy. According to the WHO's task force on strategic plan set to achieve a target of DR-TB treatment success of 75% at the end of 2015, a number of countries, including Ethiopia, have not yet achieved the milestone. Presently, for drug-susceptible and drug-resistant TB combined, the target level of >90% is set for 2025 at the latest as one of the top 10 indicators for monitoring implementation of the End TB Strategy at global and national level [11,12].

A review of research on DR-TB treatment completion shows that despite the importance of treatment completion to the success of therapy in patients with DR-TB, little is known about barriers related to treatment completion in Ethiopian patients [12–14]. In this regard, a good understanding of the views and experiences of previously treated patients on barriers to treatment completion is crucial to improving the treatment success of DR-TB.

## 2. Materials and methods

### 2.1. Study design

A qualitative study that is explorative, descriptive and contextual in nature, was conducted from November 01, 2019 to February 30, 2020 to explore and describe the views and lived experience of previously treated DR-TB patients to gain an in-depth understanding of barriers to the successful completion of DR-TB treatment.

### 2.2. Study setting

The study settings were four DR-TB treatment-initiating hospitals in Ethiopia. Saint Peter TB Specialised and Alert hospitals are located in Addis Ababa, while Adama Referral and Bishoftu

General Hospitals are located in Oromia Region in Adama and Bishoftu towns, respectively. Hospitals were selected based on their DR-TB treatment service duration and patient numbers.

### 2.3. Population of the study

Previously-treated DR-TB patients who were willing to provide informed consent to participate in the study and who were 18 years of age and older were included in the study. Purposive sampling was used to recruit study participants, with the head nurses assisting in the process.

### 2.4. Data collection and management

During focus group discussions (FGDs), open-ended probing questions were utilized to probe the discussions. A written interview guide, which is series of questions used to guide the interview, was utilized.

Focus group interviews were conducted with 42 purposively selected DR-TB patients with previous treatment history and currently on retreatment regimen. A group of seven patients was assembled for a discussion in each focus group session. Totally six FGD sessions were conducted. The FDGs were conducted at weekends for convenience of the participants.

The researcher started each discussion session with a short introduction of the topic, purpose, and ethical aspects of the study. In addition, the researcher explained the need to audio-record the discussions to ensure that they were captured accurately. As a means of stimulating the sessions, the informants introduced themselves. An informed consent form was provided, and permission was sought from the participants.

The researcher assistant (the moderator) guides the discussion according to a written set of questions in the interview guide. The discussions were tape recorded and notes were taken in the meantime. On average the discussion sessions took around 1 hour and 30 minutes. The FGDs were conducted in Amharic, the official language of Ethiopia. The FGDs were then transcribed verbatim by the researcher and an experienced public health expert while listening to the original audio content. Subsequently, transcripts in the Amharic version were translated into English with the consultation of an experienced English translator.

### 2.5. Data analysis

Qualitative content analysis is the analysis of the content of narrative data to identify prominent themes and patterns among the themes [15]. In this study, a content analysis approach was employed to describe and explore the lived experience of DR-TB patients on barriers to treatment completion of DR-TB treatment. All aspects of data management and analysis were performed by using ATLAS.ti version 8 software application. In order to verify the findings of the qualitative data set, a professional qualitative research expert was utilized as a co-coder in the process.

### 2.6. Ethical approval and consent to participate

Ethical clearance was sought from the Ethics and Higher Degrees Committee of the Department of Health Studies in the College of Human Sciences at the University of South Africa (HSHDC/834/2018, February 7, 2018).

Informed consent was sought by giving adequate information about the study and an explanation regarding their free choice to consent or decline participation voluntarily. A consent form was prepared, which needed to be signed before the FGDs were commenced. Finally, written consent was obtained after being signed by participants. Confidentiality was protected through appropriate procedure implementation.

## 3. Results

The mean age of the FGD participants was 31 years, ranging from the youngest aged 19 to the oldest 56 years. The majority of the discussants 15 (35.7%) were in the 25–34 age group. 30 (71.4%) of the participants were males. The majority of the participants, 33 (78.6%) were from urban areas by residential address. The majority of the participants were single 26 (61.9%) and unemployed 34 (81.0%). Table 1 depicts information of the participants included in the FGDs.

From the FGDs with previously treated DR-TB patients regarding their lived experience and perceived barriers to treatment completion, seven themes emerged, which comprise: (1) drug-related factors; (2) clinical issues; (3) psycho-emotional challenges; (4) socio-economic challenges; (5) service related issues; (6) patient-related factors; and (7) provider-related factors. Table 2 illustrates themes and subthemes emerged from data analysis.

### 3.1. *Theme 1: Drug-related factors*

This theme explores drug-related factors affecting successful treatment completion of previously treated DR-TB patients. The theme comprises three sub-themes which included adverse events, high pill burden, and long treatment duration.

#### 3.1.1. *Sub-theme 1.1. Adverse events.* Some of the focus group discussants explained that the second-line anti-TB drugs cause untoward adverse events that affect patients' smooth treatment adherence and completion. The following experiences were mentioned by the FGD participants:

> "It crushes my body. I get tired after taking medicine. I couldn't do anything after that. It makes me sleepy. I will be better after I eat some food and drink some milk. Otherwise, the symptoms may last for 7 to 8 to 10 hours" (M-FGD6).

> "The drug is heavy. Sometimes there is a headache. Most of the time, it makes me vomit" (F-FGD6).

Some of the FGD participants described how the discomfort of the second-line anti-TB medicines has been one of the perceived barriers to treatment completion. They have experienced

**Table 1. Demographic detail of participants of the focus group discussions (N = 42).**

| Variables | Category | Frequency | % |
|---|---|---|---|
| 1. Age categories | 18–24 | 12 | 28.6 |
| | 25–34 | 15 | 35.7 |
| | 35–44 | 10 | 23.8 |
| | 45–54 | 4 | 9.5 |
| | 55–65 | 1 | 2.4 |
| 2. Gender | Male | 30 | 71.4 |
| | Female | 12 | 28.6 |
| 3. Address | Urban | 33 | 78.6 |
| | Rural | 9 | 21.4 |
| 4. Marital status | Married | 12 | 28.6 |
| | Single | 26 | 61.9 |
| | Divorced | 3 | 7.1 |
| | Widowed | 1 | 2.4 |
| 5. Employment | Employed | 8 | 19.0 |
| | Unemployed | 34 | 81.0 |

**Table 2. Perceived barriers to treatment completion-themes and sub-themes.**

| Themes | Sub-themes |
|---|---|
| 1. Drug-related factors | 1.1. Adverse events |
| | 1.2. High pill burden |
| | 1.3. Long treatment duration |
| 2. Clinical issues | 2.1. Consulting after long illness |
| | 2.2. Delay in diagnosis |
| | 2.3. Sub-optimal dose |
| 3. Psycho-emotional challenges | 3.1. Emotional trauma |
| | 3.2. Community stigma |
| | 3.3. Health professional stigma |
| 4. Socio-economic challenges | 4.1. Economic challenges |
| | 4.2. Lack of social support |
| 5. Service-related issues | 5.1. Interruption of monitoring tests |
| | 5.2. Weak follow-up |
| | 5.3. Accessibility |
| 6. Patient-related factors | 6.1. False sense of recovery |
| | 6.2. Returning to habits already quitted |
| 7. Provider-related factors | 7.1. Lack of responsiveness |
| | 7.2. Poor communication |

pain, vomiting, and discomfort while taking the drugs. In order to not feel the pain, the patient could quit the medication. The following statements were given by the FGD participants about their own experiences of discomfort due to the drugs:

> "It makes you vomit every time you take the medicine, which is very painful, so you may feel like quitting the drug in order to prevent that pain coming" (M-FGD3).

> "… let alone two years, I did not think I would swallow for four months. When you swallow the pill, it stinks. The discomfort begins here from your throat" (M-FGD1).

**3.1.2. *Sub-theme 1.2. High pill burden.*** This sub-theme refers to previously treated DR-TB patients' lived experiences of the high pill burden of second-line anti-TB drugs as their perceived barrier to treatment completion during their previous treatments. One of the FGD participants stated the following:

> "Fifteen to sixteen pills are taken at a time. People may become frustrated and quit treatment" (M-FGD1).

Another participant reiterated.

> "You know now you are struggling when you swallow sixteen to twenty pills. It has features of nausea and vomiting" (M-FGD4).

Another participant reiterated that too much pill burden could cause loss of hope and result in quitting the therapy.

> "The medicine is too much, because of that you lose hope" (F-FGD5).

**3.1.3. *Sub-theme 1.3. Long treatment duration.*** The treatment duration of DR-TB with a second-line anti-TB treatment regimen takes longer time, and it was mentioned in the discussion that the long treatment duration was one of the factors for discontinuing the treatment. During the discussion, the long duration was boring and could cause loss of hope in patients. The following statements were mentioned by the FGD participants during the discussion:

> "The other thing is that the duration of the treatment is long and boring" (M-FGD4).

> "The drug is taken for a long time, so that as time passes you lose hope in everything. And there are people who run their own businesses before they become sick. So when you are taking the drugs, everything stops. You know that you will not create or get that business again, so you lose hope. Therefore, when a person loses hope, he/she will not have an interest in taking the medicine and getting cured again" (M-FGD3).

## 3.2. *Theme 2: Clinical issues*

This theme discusses clinical issues affecting successful treatment completion of previously treated DR-TB patients. Under this theme, there are three sub-themes which included consulting after a long illness, delay in diagnosis, and sub-optimal dose.

**3.2.1. *Sub-theme 2.1. Consulting after a long illness.*** The discussants explained that consulting after a long illness when the disease becomes serious, causing harm to their lungs, might hinder successful treatment completion and cure. The following were verbatim statements given by the focus group discussants:

> "They come for treatment after a long time without laboratory investigation and after the disease has become serious and has already caused harm to their body" (M-FGD3),

> "My first TB was fluid in my lungs [pulmonary effusion], and fluid was removed from my lungs two times. My lung accumulated fluid for the third time. After that, I started MDR treatment, but till now … I have not felt good" (M-FGD3).

**3.2.2. *Sub-theme 2.2. Delay in diagnosis.*** This sub-theme refers to FGD participants' experience of their delay in diagnosis of DR-TB, resulting in delayed clinical decisions in their treatment journey, complicating their successful treatment completion.
One participant stated:

> "At the health centre, they treated me for more than two months without diagnosing my real problem. They said typhoid or typhus. There is a delay in diagnosis. They give you painkillers. The pain may stop, but inside the disease continues to hurt you" (M-FGD1).

Another participant explained her experience of the long delay in her DR-TB diagnosis before she was actually diagnosed after more than a year of trial.

> "I was living in Wollega province and got sick there. My family did not take it seriously and took me to a nearby health facility. There, they ordered me tablets and injections. But my illness did not resolve; rather it was getting worse. Then they took me to Ambo and I had an x-ray examination. I was ordered six months of treatment for intestinal TB. On the tenth day of starting treatment, my belly swelled like a pregnant woman. It prevented me from any movement. I couldn't stand up from my bed. I was unable to eat or drink. Really, I was near death.

My family took me here to Addis Ababa, and again I was x-rayed. They said that I have fluid in my abdomen and need an operation. I was operated on at Yekatit 12 hospital. I finished six months of treatment but nothing improved. They checked me and said that the disease hurt me and that I should take a two-year treatment. Then I started MDR-TB treatment here at this hospital. I also had two more operations, and I am now feeling better" (F-FGD1).

**3.2.3. *Sub-theme 2.3. Sub-optimal dose.*** Most of the FGD participants described that one of the most common side-effects of anti-TB drugs is vomiting. Not replacing the vomited dose of the medication results in a suboptimal dose. This prevents the successfulness of the treatment and results in unfavourable treatment outcomes. The FGDs participants stated their experience as follows:

"The drugs come out immediately after swallowing. When it comes out, it should be swallowed after a while. But some patients leave it without replacing the vomited dose. This will harm the treatment and the disease will get worse" (M-FGD1).

"The drug is very hard to take. I vomited frequently for the first six months. I was admitted here for three months. When I take a drug, the immediate side-effect is vomiting. While I was admitted here, if I vomited the drug, it was replaced and I took it after 30 minutes. But after discharge, the drug was given to me from the health centre, and there is no replacement. Because my home and the health centre were far away, there was nothing to be replaced" (F-FGD3).

### 3.3. *Theme 3: Psycho-emotional challenges*

This theme refers to study participants' lived experiences regarding emotional and psychological challenges that affected their successful treatment completion.

**3.3.1. *Sub-theme 3.1. Emotional trauma.*** Most of FGD participants stated that emotional trauma, feelings of loneliness, and at times, loss of hope were making their treatment process painful. The following were psychological challenges expressed by FGD participants:

"My reason for interruption was as follows…. When I was diagnosed with TB, I was isolated. Everyone just avoided me. All those who knew me in my neighbourhood ignored me, even to the point of spending the entire day alone in the forest. I got to the point of committing suicide. Many things happened to me. It is even to the point of eating out of a restaurant's trash" (M-FGD2).

"At the beginning, I was a waitress. Now I couldn't do that. When my own father knew I was sick with TB, he went so far as to deny it to me, saying that he did not know me. There is no family besides mine. No family near me. Except for God, there is no one around me" (F-FGD2).

**3.3.2. *Sub-theme 3.2. Community stigma.*** FGD participants explained their experiences of stigma and discrimination they encountered from the community while on their DR-TB treatment. The stigma and discrimination they faced were so bad that they were forced to discontinue their treatments. The focus group participants explained the stigma and discrimination associated with the treatment of DR-TB as follows:

"Primarily, the reason I stopped the drug was because of discrimination. For example, when we wear a mask and enter the gate, starting from a health professional, they keep

a distance from you of around eight meters. When we walk down the road, everybody changes their ways on opposite sides of you. When everybody referred to us as MDR" (F-FGD2).

"The stigma and discrimination is so hard that I lost my friends with whom we grew up together and many people because of this disease. I'll tell you frankly, I prefer HIV to this disease" (M-FGD3).

**3.3.3. Sub-theme 3.3. Health professional stigma.** In addition, denial of service by health professionals due to stigma was mentioned by participants of the focus groups. The following were verbatim statements by the study participants:

"The service is hard for TB patients. You cannot get the service like other patients directly. You have to wait until 10 minutes before 6 o'clock to get the laboratory service. [This is to give space for other patients to leave the area]. And sometimes you do not get them and get the service from outside [private labs]. The laboratory and X-ray services are not comfortable for TB patients" (M-FGD5).

"I am from …. The doctors over there discriminated against me and were not interested in treating me. I was diagnosed here, got my drug, and went back there, but I could not get anyone to receive and treat me. So I came back here, and they rented me a room, and now people are helping me for living" (M-FGD4).

### 3.4. *Theme 4: Socio-economic challenges*

This theme refers to the experiences of study participants in the focus groups about lack of social support and economic burdens they faced while on treatment, forcing them to the extent of discontinuing their treatment.

**3.4.1. Sub-theme 4.1. Economic challenges.** Most of the FGD participants indicated that they experienced economic burdens while on treatment, to the level of being unable to support their livelihood and being forced to quit their therapy before completing it.

Some of the participants explained their experience of financial strains they faced while on treatment as follows.

"Taking the drugs outside the hospital, being discharged from inpatient, is impossible for many patients. Everyone has no income. Most of us don't have a home. At that time, a patient can stop the treatment. Here in the hospital, the treatment is good. But after six or seven months when you are discharged, it is very difficult for most of us. While in the hospital, I have seen many patients return back to the hospital after interruption of treatment. When I heard their reasons, I was shocked. First, they have nothing to eat once discharged and go out of the hospital. Second, they don't have a residence or home" (M-FGD1).

"Life is convenient here for four to five months, but when you are discharged, everything becomes new to you. Life is also not convenient [after discharge]. You can't afford to rent a house. So, you are vulnerable to quitting the treatment. You are worried about what you drink or eat; about what you do and then swallow the drugs" (M-FGD4).

Furthermore, patients on the DR-TB treatment lose their jobs because of their illness and side-effects of the drugs and do not have any source of income to buy at least some of the nutritious food available locally, and they also cannot support their dependents at home.

These economic challenges make continuing the treatment difficult for them. Participants noted how loss of a job brought consequences during treatment as follows:

> "The main problem is the economic problem. The drugs need nutritious food, otherwise they cause gastric pain and the like. It's difficult to take one year and nine months with this condition. It is difficult to continue using drugs if there is no work or other source of income" (M-FGD4).

> "Challenges, there are many challenges. The main challenge for people to interrupt treatment is the economic challenge. Once you are admitted, you lose your job. You don't have any source of income" (M-FGD1).

**3.4.2. *Sub-theme 4.2. Lack of social support.*** This sub-theme discusses lack of social support such as material support (food, financial incentives, transport fees, living allowance, and housing incentives) that addresses the indirect costs incurred by patients or their attendants to access health services and, possibly, tries to mitigate the consequences of income loss related to the disease.

In the treatment process, patients need regular material support, for instance in the form of adequate monthly food baskets, covering house rents and subsistence for the dependent family members, including children, since patients lose jobs or cannot work while on treatment. This can oblige patients to look for work or daily labour for the purposes of their livelihoods, and then discontinue their treatment. Participants described challenges during treatment as emanating from the lack of material support they faced, as follows:

> "The other is related to life. For example, I rent a house and I learn. So I have to pay for both. In the meantime, I have no one to support me. So, there are many things that stress you and make you want to discontinue the drug" (M-FGD6).

> "When there is nobody to support you, You might decide to stop the medicine and go to work in order to live and make something that you think of" (F-FGD3).

One of the challenges for patients during treatment follow-up was the problem of transportation. The treatment requires daily observation at follow-up centres and monthly follow-ups at treatment initiating centres. Therefore, transportation is the backbone of social support that should be in place during the whole course of the treatment regimens. However, transportation support was mentioned to be one of the challenges for patients affecting their successful treatment completion. The following statements were verbatim explanations of the study participants:

> "… to come and collect the drug, there is no money for transport. … I remember, one day, I had no money to come here to collect … I came from my home to Merkato on foot, then asked a stranger on the street for some money for the taxi and came here" (M-FGD2).

> "You know, there is a problem with transportation. I have difficulty coming here from my home most of the time. So, I used to come intermittently" (M-FGD3).

### 3.5. *Theme 5: Service-related issues*

This theme refers to previously treated patients describing how unavailability and interruption of laboratory services and weak patient follow-ups affected clinical decisions and, consequently, medical conditions getting worse for them.

**3.5.1. *Sub-theme 5.1. Interruption of monitoring tests*.** Some of the FGD participants indicated that there were interruptions of laboratory monitoring tests during treatment follow-up periods, making timeous monitoring and management of drug adverse reactions impossible in case one occurred. If adverse events are not managed timeously, patients may get frustrated and discontinue their treatment. On top of that, adverse events can also lead to bad treatment outcomes to the extent of taking life.

"I get from outside [laboratory monitoring tests]. They send you to …. So it is good if necessary laboratory tests are done here. As we know, the drugs have so many side-effects and they affect the kidneys, eyes, ears, and liver. If they do check-ups every month or at least every 3 months, I think it is good" (M-FGD6).

"Previously they did blood investigations, but currently it is not done" (M-FGD6).

Some of the FGD participants further stated the unavailability of the laboratory services ordered by their physicians during treatment with second line drugs.

"I did not get service here. I have got the service from outside, at private laboratories. They do not have the service here" (M-FGD6).

"It is also good if laboratory tests, which are sent outside and done in private labs, are performed here in the hospital. Because we get an immediate solution if it is done here. If a patient does not have money at hand at that time, it takes time till he gets that money and clinical decisions are not made on time. So, things get worse for the patient" (M-FGD1).

**3.5.2. *Sub-theme 5.2. Weak follow-up*.** Treatment follow-up at treatment initiating centres (TICs) is usually conducted every month, and monitoring of adverse events and treatment responses is systematically followed and recorded. FGD participants' experiences indicated that treatment follow-ups were done every three months in some TICs. As a result, any patients' concerns, both clinical and social, could not be addressed timeously and patients may interrupt their treatment in such conditions.

"I used to give blood tests every 3 months" (F-FGD5).

"I had a check-up every 3 months, but I was taking the medicine monthly…" (F-FGD4).

**3.5.3. *Sub-theme 5.3. Accessibility*.** As explained by the FGD participants, many health facility locations were far from their home town or districts. They were forced to rent a house and sustain the high cost of transportation to continue their follow-ups. The FGD participants stated accessibility issues as follows:

"There is no health centre in my home town, and I should stay there to follow my treatment. I had to rent a house. I had to prepare my own food. There is a transport cost. Those are problems I faced. As it is known, the rent of the house is very expensive. On top of that, I do not have a job" (M-FGD3).

"The second reason is distance from home to treatment site, which can also make a patient quit. I struggled to come here while on follow-up because it is far from my home" (M-FGD6).

## 3.6. *Theme 6: Patient-related factors*

This theme refers to patient factors affecting the successful treatment completion of previously treated DR-TB patients.

**3.6.1.** *Sub-theme 6.1. False sense of recovery.* Some FGD participants described that when patients take medication for a couple of months and symptoms start to subside, patients could develop a false sense of cure and quit their therapy, in spite of the recommendation that they should continue medication for the prescribed duration.

"I completed the first TB treatment. However, I did not have the final check-up. Just because I was recovered at the time, I just went out and left. Then it came back again…." (M-FGD2).

"And when you feel good and feel recovered and think you brought change; you might think that it is enough to take the medicine" (M-FGD3).

**3.6.2.** *Sub-theme 6.2. Returning to habits already quitted.* This sub-theme refers to previously treated patients' experiences of returning to their old habits and addictions. Consequently, this affects treatment adherence and eventually causes treatment interruption and unsuccessful outcomes.

"Of course, you know, sometimes the drugs make you emotional and patients may start the habits they have already quit, like smoking and drinking alcohol. This exacerbates their health problems" (M-FGD1).

"On the other hand, I know patients who felt improvement and returned to their addiction. The drug brings harm when you start to use it for addictive things before completion of treatment" (M-FGD6).

### 3.7. *Theme 7: Provider-related factors*

This theme refers to the experiences of the FGD participants regarding healthcare providers' responsiveness during times of need and the establishment of satisfactory communication during the treatment period.

**3.7.1.** *Sub-theme 7.1. Lack of responsiveness.* Most of the study participants stated that responsiveness was not in place at inpatient wards during their stay in the hospitals, especially during emergency situations. The response of the ward team was not fast enough to reverse situations. Most of the patients stated the following:

"… If I say I am feeling sick, they say drink milk, apply this ointment, or drink water. They even did not give us the time you gave to me now. No one talked to us; so, what is the meaning of our stay in the hospital? This is the problem…." (M-FGD3).

"Here in… when we get sick, feel dizzy or suffer convulsions, they only say it is okay to drink water or something. There is no one who is responsible for giving us appropriate treatment. I have seen patients in distress and affected when they waste time trying to call doctors in an emergency" (F-FGD4).

**3.7.2.** *Sub-theme 7.2. Poor communication.* This sub-theme involves study participants describing an experience where there was poor communication from healthcare providers in telling them the seriousness of their problems and explaining their results. While some spoke, the others nodded in agreement, and the following were some of their verbatim vignettes:

"They did not tell us the result and it affects our psychology. We lose hope and if they tell us nothing about the result, we think that they do not have hope for us and that there is something worse" (M-FGD3).

"Then after my body started to get rigid and rigid. No one communicates to us properly" (F-FGD2).

Adequate information was crucial and should be part of the counselling process so that patients would have sufficient awareness to deal with their medication and treatment. Not giving adequate information was mentioned as being one of the challenges during the treatment process by the participants. The following points were mentioned during the discussion by the study participants:

"No one gives adequate information on how to take medicine. There are 14 tablets given to all patients, and no one tells you that this drug is for this symptom or for that" (M-FGD4).

"As to me, it is not enough. Because, at least, no one is telling you how to take your drugs. For example, if the drug has to be taken within 30 to 40 minutes, patients take it the whole day because they do not know how to take" (M-FGD5).

## 4. Discussion

This study explored previously treated DR-TB patients lived experience concerning barriers to successful treatment completion. Discomfort due to the second line anti-TB drugs are common and at a time may hinder successful treatment completion. It has been revealed, in qualitative study done by McNally et al. to explore the experiences and perceptions of DR-TB patients and healthcare providers (HCPs) in Loreto in order to identify barriers and facilitators to achieving optimal outcomes, that only the sight of the pills was enough to trigger nausea and vomiting [4]. Moreover, it is explained that the nausea which appear right after swallowing of the tablets was incapacitating to patients [16]. Furthermore, it has been reported, in a study conducted in Ethiopia, that discomfort due to nausea and vomiting, and pain occurred to muscle and joint forced patients to interrupt and quit their treatment before completion [17].

A number of studies have indicated that treatment of DR-TB is challenging and complicated, takes long duration, and is associated with frequent and several adverse events [3,18–22]. In fact, many studies established the fact that adverse events of second line anti-TB drugs are one of the factors associated with unsuccessful treatment completion. Moreover, Jakasania et al. stated that while on DR-TB treatment, adverse events are one of the leading causes of unsuccessful treatment outcomes among patients [18]. In addition, in the qualitative study by Shringarpure et al., adverse events were pointed out as one of a major barrier to adherence in the long and difficult journey of DR-TB treatment [23].

It was revealed in the current study that the high pill burden and the long treatment duration were frustrating, boring, bringing loss of hope in patients and may result in quitting therapy. A number of studies conducted locally and globally confirm the situation. Eshetie et al. and Molie et al. argue that long treatment duration could compromise treatment adherence which remains a significant challenge in achieving successful treatment outcomes in Ethiopia [9,24]. Similarly, it was also reported that one of the themes identified for reasons for lost-to-follow-up is struggle with prolonged treatment [23].

This study revealed that, consulting after long illness (after repeated first line treatments, after attempting treatment by traditional healers) when the disease become serious causing harm to their lungs, might hinder successful treatment completion. This creates suitable lodging environment for the bacteria and cavities which can make the treatment difficult. Further, repeated attack of the lung result in fibrosis [scarring] which is difficult for the drugs to penetrate the scar tissue and make a cure difficult [3,5,9,25].

FGD participants lived experience showed delay in diagnosis of DR-TB as one of the factors related to unsuccessful treatment outcomes. Similar findings were identified in studies conducted in Zimbabwe by Matambo et al., in Myanmar by Htun et al., as well as in Armenia by Khachatryan et al., where it was noted that delay in the process of diagnosis and initiation of treatment may have considerable impact on disease progression and prognosis, finally contributing to unsuccessful treatment outcomes [26–28].

The current study described that one of the most common side-effects of anti-TB drugs is vomiting. Not replacing the vomited dose of the medication results in a suboptimal dose. Consistent to our findings, in a study set out to analyse adverse drug reactions and treatment outcomes of DR-TB, Dela, Tank, Singh and Piparva explained that adverse drug reactions associated with second line drugs, resulting in insufficient treatment, and were significantly associated with non-treatment adherence and defaulter outcome, thereby affecting successful treatment completion [29].

This study also revealed that previously treated DR-TB patients experienced psycho-emotional challenges affecting their treatment adherence and successful treatment completion. Consistent with our findings, different studies have reported that patients on DR-TB treatment may commonly experience the symptoms of psycho-emotional trauma, including stigma, discrimination, isolation, helplessness, and loneliness [4,21,30]. Furthermore, in another study it is reported that this psycho-emotional trauma is one of the factors associated with TB treatment non-adherence, which also coincides with our findings [8].

A study done in Ethiopia that reported economic problems are associated with as well as independently predict treatment non-adherence. In addition, it was also noted that patients who have financial problems suffer extra costs incurred to cover transportation and living costs in the long treatment duration [8]. Furthermore, it was reported in another Peruvian study, poor adherence is one of the barriers to successful treatment outcomes [4].

Similar to our findings, many studies showed that social problems could negatively affect DR-TB treatment outcomes. Study conducted in Nepal by Baral et al. stated that DR-TB treatment causes social problems [31]. In addition, qualitative study done by Shringarpure et al. revealed that social problems are major barriers to treatment adherence and continuous engagement in care [23]. In summary, the socioeconomic impact of DR-TB treatment including extra costs for accommodation and food, loss of employment, loss of family and social network support and isolation negatively influences treatment adherence and successful outcomes [4,8].

Baseline and routine monitoring laboratory tests are required for patients on treatment with second-line anti-TB drugs [32]. Moreover, laboratory tests can detect occult adverse events that cannot be noted by the patient or HCPs [20]. At the same time, in resource limited settings, where availability of regular supply of monitoring tests is inadequate, identification and management of adverse events cannot be done as required [33]. A study shows that in Ethiopia, there is deficiency of constant supply of laboratory reagents in the public sector periodically [34].

Regular clinical follow-up as well as timeously monitoring and management of adverse events are among the basic principles of DR-TB treatment. Sabur et al. stipulated that regular patient follow-up and monitoring decrease occurrence of adverse events [35]. In addition, regular patient follow-ups give a chance to educate patients about the use of each medication and possible adverse events so that patient could report during clinical days, there by successful treatment completion take place eventually [22].

In the current study, accessibility was found to be one of the underlying reasons for treatment interruption. This finding was in line with studies conducted by McNally et al., Jakasania et al., and Benbaba, Isaakidis, Das, Jadhav, Reid and Furin that reported accessibility

of care to be one of the important factors for adherence to treatment and better compliance [4,16,18]. In addition, the WHO recommends that health systems should endeavour to guarantee accessible DR-TB treatment service corresponding to need of patients and also reiterates the need of addressing significant barriers to accessing DR-TB care like distance to travel [6].

This study revealed that patients may develop false sense of cure and could quit their therapy, in spite of the fact that they should continue medication for the prescribed duration. In this regard, the results of qualitative study done by qualitative study done by Shringarpure et al. affirms the situation. Thus, it was reported that patients were prone to quit treatment upon feeling resolution of symptoms and improvement of physical signs [23].

Different scholars have noted that addiction has been the bottle neck for smooth treatment completion by affecting adherence and initiating lost-to-follow-up. It is reported that addiction to khat, tobacco, alcohol or cannabis is common in patients compromising their capacity of adherence to the prescribed regimen till end of therapy. In addition, it was also indicated that substance abuse was associated with defaulting from treatment and one of the reasons for not returning for treatment by those already defaulted [23]. Furthermore, Matambo et al. connoted that substance use has been found to have association with poor treatment outcomes [26].

A trustful relationship between health care providers and patients is important to establish good communication, to gain patients' trust and confidence in the treatment process [4]. Likewise, Bhering et al. noted that a relationship based on trust with emotional support produces improved adherence and successful treatment completion [21]. On the contrary, it was reported that relationship based on hierarchy, where patients were uncomfortable to express their regards, negatively affected treatment adherence [4]. Moreover, in a qualitative study to understand patients' and providers' perspectives on reasons for lost-to-follow-up in India reported that, on patients perspective, engagement in unfriendly relationship with health providers and feeling not getting information about one's medical reports and health status progress, resulted in treatment interruption and lost-to-follow-up [23].

## 5. Limitations

Recall bias: previously treated patients are likely to have forgotten their past treatment experiences and may not have shared adequate information about their previous treatments.

## 6. Conclusion

This study has provided an understanding of the views and lived experiences of previously treated DR-TB patients on barriers to successful treatment completion. The barriers identified include drug-related factors, clinical issues, psycho-emotional challenges, socio-economic challenges, and service-related issues, as well as patient- and provider-related factors. Addressing the above factors, which pose barriers to the successful completion of DR-TB treatment, demands policy-level decisions and programmatic guidance at the national level based on best practices, as well as good programmatic implementation from actors through regional and health facility-level structures.

To enhance successful treatment completion of DR-TB patients, several recommendations are proposed. Firstly, it is essential to ensure proper clinical and laboratory monitoring, alongside routine screening for common comorbidities throughout the treatment process. Access to DST and culture should be improved by expanding decentralized laboratory services and revitalizing specimen referral systems. Continuous counseling and need-based psychosocial support are crucial elements for patients undergoing treatment. Sensitization efforts targeting families, communities, and employers can significantly enhance awareness of DR-TB.

Interactive sensitization programs should be organized to combat the stigma associated with the disease. Initiating community-based DR-TB care principles can improve accessibility and support networks. Mental health care integration within the PMDT is vital at both policy and implementation levels. Establishing a specialized surgical treatment center and ensuring high-quality respiratory care evaluation for patients with advanced disease is essential. Finally, expanding infrastructure and services for DR-TB care to cover wider geographic areas within the country is paramount for comprehensive management and successful treatment completion.

## Supporting information

**S1 Checklist. Inclusivity in global research.**
(DOCX)

## Acknowledgments

We would like to thank the Federal Ministry of Health for facilitating our study and the participants for their unreserved effort to participate in the study.

## Author contributions

**Conceptualization:** Ahmed Reshid Tusho.

**Data curation:** Ahmed Reshid Tusho.

**Formal analysis:** Ahmed Reshid Tusho.

**Investigation:** Ahmed Reshid Tusho.

**Methodology:** Ahmed Reshid Tusho.

**Project administration:** Ahmed Reshid Tusho.

**Resources:** Ahmed Reshid Tusho.

**Software:** Ahmed Reshid Tusho.

**Supervision:** Sheila Theresa Mokoboto-Zwane.

**Validation:** Sheila Theresa Mokoboto-Zwane.

**Writing – original draft:** Ahmed Reshid Tusho.

**Writing – review & editing:** Sheila Theresa Mokoboto-Zwane.

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
