## [Decision Letter · Decision Letter 0]

9 Jan 2024

PGPH-D-23-02426

Treatment completion of drug-resistant tuberculosis in Ethiopia: a perspective from healthcare users

Dear Dr. Tusho,

Thank you for submitting your manuscript to PLOS Global Public Health. After careful consideration, we feel that it has merit but does not fully meet PLOS Global Public Health’s publication criteria as it currently stands. Therefore, we invite you to submit a revised version of the manuscript that addresses the points raised during the review process.

This is an important study about the perspective of healthcare providers towards DR-TB treatment completion. Please address the reviewers' comments before resubmission; focus specifically on reviewer 2 and 3 if there are any conflicts. 

We look forward to receiving your revised manuscript.

Kind regards,

Mareli Misha Claassens

Guest Editor

Journal Requirements:

1. Please include a complete copy of PLOS’ questionnaire on inclusivity in global research in your revised manuscript. Our policy for research in this area aims to improve transparency in the reporting of research performed outside of researchers’ own country or community. The policy applies to researchers who have travelled to a different country to conduct research, research with Indigenous populations or their lands, and research on cultural artefacts. The questionnaire can also be requested at the journal’s discretion for any other submissions, even if these conditions are not met.  Please find more information on the policy and a link to download a blank copy of the questionnaire here: https://journals.plos.org/globalpublichealth/s/best-practices-in-research-reporting . Please upload a completed version of your questionnaire as Supporting Information when you resubmit your manuscript.

2. Please send a completed 'Competing Interests' statement, including any COIs declared by your co-authors. If you have no competing interests to declare, please state "The authors have declared that no competing interests exist".

3. Please amend your detailed Financial Disclosure statement. This is published with the article. It must therefore be completed in full sentences and contain the exact wording you wish to be published.

If you did not receive any funding for this study, please simply state: “The authors received no specific funding for this work.”"

4. In the online submission form, you indicated that "date can be accessed according to the PLOS Data Policy". 

Additional Editor Comments (if provided):

Reviewers' comments:

Reviewer's Responses to Questions

**Comments to the Author**

1. Does this manuscript meet PLOS Global Public Health’s publication criteria ? Is the manuscript technically sound, and do the data support the conclusions? The manuscript must describe methodologically and ethically rigorous research with conclusions that are appropriately drawn based on the data presented.

Reviewer #1: Yes

Reviewer #2: Partly

Reviewer #3: Partly

2. Has the statistical analysis been performed appropriately and rigorously?

Reviewer #1: Yes

Reviewer #2: N/A

Reviewer #3: N/A

3. Have the authors made all data underlying the findings in their manuscript fully available (please refer to the Data Availability Statement at the start of the manuscript PDF file)?

Reviewer #1: Yes

Reviewer #2: No

Reviewer #3: No

4. Is the manuscript presented in an intelligible fashion and written in standard English?

Reviewer #1: Yes

Reviewer #2: No

Reviewer #3: No

5. Review Comments to the Author

Reviewer #1: Comments

The authors aimed to provide the perspectives of healthcare users regarding treatment completion of drug-resistant TB and the barriers that affect the outcome "The aim of this study was to explore and describe the views and lived experiences of previously treated drug-resistant tuberculosis patients on barriers to successful treatment completion."

From the reviewer perspective the authors have achieved the aim of the study. The paper is concise and reported the method and results well.

Minor concerns

1. The paper contains instances whereby the authors mention studies (plural) and only cites one source.

2. The authors did not identify the limitations of the study.

3. The authors do not provide further information on the semi-structured interview, how did the authors develop the structure of the interviews? Is there information on this?

4. In the discussion section the authors link the results to previous research well, it might be considered to add a sentence or two on practical ways to address the barriers highlighted in the study.

5. The reference list contains up-to-date citations, however it is not clear to the reviewer if the formatting of the references are accurate.

6. In some instances the font styles and spacing are not uniform.

Reviewer #2: Overall

The authors provide an overview of the experiences of people who previously received MDR-TB treatment at different health facilities in Ethiopia. The authors present seven key barriers to treatment adherence/completion and describe data from participants who participated in group discussions. While the manuscript provides important information that could be valuable in helping patients at risk of treatment disruption, there are several shortcomings that the authors need to address.

Firstly, the multiple sub themes and subheadings is quite difficult to follow. It would be worth simplifying the structure of the results instead of breaking down into 18 (!) subheadings. Perhaps reduce to (for instance) 5 major themes as there is some overlap between several of the themes and subthemes. The alternative is to focus on some of the barriers. The data is rich and there is much to be gained in publishing the data, however, the current structure is not very digestible. In addition, while barriers are important, it seems that these participants might have much to contribute in terms of identifying facilitators (for instance, described in subtheme 3.3).

I would recommend a restricting of the manuscript, being more mindful of the types of barriers, including which ones are amenable and how facilitators can be identified.

The manuscript will also benefit from a more detailed description of the participants. Are these participants who experienced treatment interruptions? Where they mostly adherent? How were these participants recruited and when were they previously treated for TB? How many participants have had repeat episodes of TB? Do they have other comorbidities? Do they have other household members diagnosed with TB? All of these factors are likely to impact their lived experience of TB treatment.

The analysis process is also not adequately described. The authors refer to content analysis but the way they describe their data seems to be much closer aligned to thematic analysis.

Strengths and limitations section need to be added.

The authors use MDR-TB, RR-TB and DR-TB interchangeably – each of these refer to a specific phenomenon. Please be mindful when describing and try to be consistent.

Abstract

The authors would benefit from a language edit, simplifying some of the very lengthy/verbose sentences. E.g., the fourth sentence in the abstract stretches over four lines and can be shortened/rewritten. In addition, the authors often revert to passive voice.

In the abstract, the results listed are sparse and more details are needed. While the authors set out to describe the barriers to treatment completion, it would also be interesting to note facilitators, or ways that participants overcome some of these barriers.

In the abstract, the conclusion and recommendations does not seem to match the findings. The authors suggest that national-, policy-, and facility level interventions are needed, but some of the barriers identified are interpersonal. A bit more justification is needed – or an expanded description of the findings.

Introduction

The authors provide several statements without substantiation/references. E.g., paragraph 5, sentences 1 and 2 both need references. While the third sentence does include references, it is not clear what “a number of variables” refers to?

The fourth paragraph is repetitive and can be deleted.

On page 3 of the content (top), the authors refer to “anecdotal evidence” but it is not clear where this evidence is from – their own anecdotal evidence or noted in another literature source?

Materials and Methods

- Study setting

Include some details on whether these settings are all urban/rural, formal/informal, high/low TB burden, high income/low-income settings.

- Population

How were participants recruited?

- Data collection and management

Several typos in this section.

Which if the authors were involved in data collection/analysis?

- Data analysis

This section needs several references. Content analysis is a method to quantify qualitative data. If the researchers conducted content analysis, they would need to specify their coding frame used to identify quantifiable patters to base their conclusions on. From the description of the findings, it seems like they used thematic analysis – but they would need to revise their description of the analysis process and include relevant references.

Results

Most of the participants were male – is this reflective of the TB epidemic in Ethiopia?

Marital status – was “co-habiting” or “in a relationship” offered as options?

The authors note that these are participants who were previously treated. Did all complete treatment or did some default on treatment? How many episodes have the previously had? If available, it would be worth including this information in the demographic section. Did any participants have other co-morbidities (like HIV or diabetes) which could have further complicated treatment adherence?

Throughout this section it would be helpful to see the age of the participants along with the gender when quotes are provided.

- Sub-theme 1.1

The participants mentioned that they thought they would not be able to adhere for an extended time because of the smell of the tablets. Did they adhere?

- Sub-theme 2.1. Consulting after a long illness

It is not clear how this theme presents a barrier to adherence. Do the authors suggest that illness severity/progression is a barrier to adherence? It also seems similar to theme 2.2. The quotations illustrate that TB is hard on the body but not how it relates to treatment adherence.

- Sub-theme 5.3. Accessibility

This section seems to be specifically related to rural/urban access to healthcare. Please indicate how many of the participants are from rural areas in your demographic section and relate it to the quotes provided in this section.

- Sub-theme 6.2. Returning to habits already quitted

The quotes provided are for perceptions of other patients. Did any participants report that they reverted to these habits?

- Sub-theme 7.1. Lack of responsiveness

This subtheme does highlight an important concern related to quality of care. However, it does not illustrate challenges with adherence (when hospitalised, would patients not be more likely to take/complete treatment – even if administered by rude staff or without adequate explanation)?

Discussion

Second paragraph: Please describe where these studies were conducted and when. There have been some improvements in available treatment for people diagnosed with MDR-TB. If these studies are older, the adverse events might have been more severe.

Third paragraph: “In line with this, the in-depth interview with health care providers indicated that DR-TB patients are not usually diagnosed timeously and come after multiple first line treatment courses” I tis unclear if this sentence (a) refers to a previous study in the literature, or (b) if the authors also conducted interviews. If (a) please make clear, if (b) it will need to be included in the methods, findings, etc.

The authors describe how side effects such as vomiting hinders adherence. However, it is not clear how sub-optimal dosing is a barrier to treatment adherence. While this is a clinical concern, it is a consequence rather than a barrier.

The term “social problems” is used but not explained. This is too broad to use without any detail.

Conclusion

The recommendation to “address these issues” is not strong enough and needs some more concrete examples/responses. How would policy makers/health workers go about addressing long treatment regimens, or adverse events, or economic challenges? While identifying the challenges is the first step, the authors do need to give it some more thought.

For some of the issues, some interventions have been implemented and the authors should recognise these (for instance, improved treatment regimens with fewer side effects, studies on treatment shortening regimens, social support interventions, etc.).

Additional comment

Language edit needed – several typos:

“.. the researcher started each discussion sessions”

“…to ensure that the researcher captures what they accurately.”

“Informed consent form provided and permission sought from the participants.”

ADR – need to write out

And repeated use of the same phrasing which could be edited:

“lived experiences”

“verbatim”

Reviewer #3: Congratulations to the authors on addressing a very important topic. I hope that you will take the time to address the suggested revisions, because these data are important to be published.

Major Comments:

1. Although the detailed quotes from participants are interesting to read, there is a lack of analysis of the context (mentioned twice in your methods that this is a 'contextual' analysis) of these quotes. There is overall, a lack of articulation of how the themes fit relative to each other, what matters more. At current it reads as a long-list of potential influences, rather than a frame through which we can really understand the challenges to adherence in this setting. What matters more to people? Vomiting? Duration? Losing friends? All these things probably all matter, but they are not all equal. Nor are they equally operable / changeable. If we seek to understand so that we can do something about the challenges, then we need to understand more.

2. The discussion is challenging. I suggest a total revision to follow a structure that flows: (a) a brief summary of what was found (high-line results), (b) a comparison of how these results compare to what is already known. At current, this is piecemeal per theme, rather than overall what we now understand about barriers to MDR-TB adherence. If one takes a step back, then what have we really learnt from this study? Did we not already know that side effects, pill burden etc. were challenges? There is also a lack of engagement with sufficient literature on the challenges of DR-TB adherence in the region and globally, with a limited number of citations. The discussion should highlight what is confirmatory and what is potentially novel from the results in a way that is concise. (c) a statement of the strengths and any limitations to extrapolation from these results.

3. The conclusion is far to bland / generic. What specific policy, practice, and future research recommendations can you make from these results? How should the care for people with DR-TB be changed in Ethiopia.

Minor Comments:

1. The abstract - is unbalanced by sub-section, more must be presented under 'results' and less under background and conclusion sub-sections. The results that are presented are only descriptive, and do not help the reader actually understand what you found. Examples from the categories, not just broad themes are necessary.

2. The first three sentences of the background are without citations. There are readily available reports on drug resistant TB epidemiology that should be cited to establish the magnitude of the issue.

3. The third and fourth paragraphs of the introduction are unnecessarily repetitive.

4. Top of page 7 - there is more than anecdotal evidence that recurrence is more likely than first-time disease episodes. See literature on post-TB lung disease.

5. Top of page 8, first two sentences are unnecessarily repetitive.

6. Under section 2.4, we do not need to read again which hospitals were part of the study.

7. Under section 2.5, please provide citations for 'a conventional content analysis'.

8. Under section 2.5, "ATLAS.ti", not "Atlas. ti"

9. First paragraph under section 3 - are the participants typical of people with DR / RR-TB locally?

10. Top of page 25 - what reagents were needed / provided?

6. PLOS authors have the option to publish the peer review history of their article (what does this mean? ). If published, this will include your full peer review and any attached files.

**Do you want your identity to be public for this peer review?** For information about this choice, including consent withdrawal, please see our Privacy Policy .

Reviewer #1: No

Reviewer #2: No

Reviewer #3: **Yes: ** Graeme Hoddinott

While revising your submission, please upload your figure files to the Preflight Analysis and Conversion Engine (PACE) digital diagnostic tool, https://pacev2.apexcovantage.com/ . PACE helps ensure that figures meet PLOS requirements. To use PACE, you must first register as a user. Registration is free. Then, login and navigate to the UPLOAD tab, where you will find detailed instructions on how to use the tool. If you encounter any issues or have any questions when using PACE, please email PLOS at figures@plos.org. Please note that Supporting Information files do not need this step.

---

## [Decision Letter · Decision Letter 1]

25 Jun 2024

PGPH-D-23-02426R1

Treatment completion of drug-resistant tuberculosis in Ethiopia: a perspective from healthcare users

Dear Dr. Tusho,

Thank you for submitting your manuscript to PLOS Global Public Health. After careful consideration, we feel that it has merit but does not fully meet PLOS Global Public Health’s publication criteria as it currently stands.

We look forward to receiving your revised manuscript.

Kind regards,

Mareli Misha Claassens

Guest Editor

Journal Requirements:

1. Please include a complete copy of PLOS’ questionnaire on inclusivity in global research in your revised manuscript. Our policy for research in this area aims to improve transparency in the reporting of research performed outside of researchers’ own country or community. The policy applies to researchers who have travelled to a different country to conduct research, research with Indigenous populations or their lands, and research on cultural artefacts. The questionnaire can also be requested at the journal’s discretion for any other submissions, even if these conditions are not met.  Please find more information on the policy and a link to download a blank copy of the questionnaire here: https://journals.plos.org/globalpublichealth/s/best-practices-in-research-reporting . Please upload a completed version of your questionnaire as Supporting Information when you resubmit your manuscript.

Additional Editor Comments (if provided):

Please have a thorough look at the second round of comments and address the comments step by step, indicating in the response letter on which pages and where exactly the changes were made.

Reviewers' comments:

Reviewer's Responses to Questions

**Comments to the Author**

1. If the authors have adequately addressed your comments raised in a previous round of review and you feel that this manuscript is now acceptable for publication, you may indicate that here to bypass the “Comments to the Author” section, enter your conflict of interest statement in the “Confidential to Editor” section, and submit your "Accept" recommendation.

Reviewer #2: (No Response)

Reviewer #3: (No Response)

2. Does this manuscript meet PLOS Global Public Health’s publication criteria ? Is the manuscript technically sound, and do the data support the conclusions? The manuscript must describe methodologically and ethically rigorous research with conclusions that are appropriately drawn based on the data presented.

Reviewer #2: Partly

Reviewer #3: Partly

3. Has the statistical analysis been performed appropriately and rigorously?

Reviewer #2: N/A

Reviewer #3: N/A

4. Have the authors made all data underlying the findings in their manuscript fully available (please refer to the Data Availability Statement at the start of the manuscript PDF file)?

Reviewer #2: Yes

Reviewer #3: No

5. Is the manuscript presented in an intelligible fashion and written in standard English?

Reviewer #2: Yes

Reviewer #3: No

6. Review Comments to the Author

Reviewer #2: While the authors have made efforts to address some of the comments, several comments were either inadequately addressed or ignored. In addition, the manuscript is still difficult to navigate with multiple sub headings and sub-themes. These could easily be combined.

Some examples below:

Comment: How were participants recruited?

This is still not addressed. Were they recruited from participant lists/folders – what does this mean for ethics? Were they recruited as they left the clinic? Were they phoned? Did nursing staff invite them to participate? Were flyers put up in the clinic? Describing that it was purposive sampling does not explain how participants were recruited.

Comment: Most of the participants were male – is this reflective of the TB epidemic in Ethiopia?

Author response: May be. According to WHO 2021 TB report, among notified TB cases case adult women were 39%, adult men were 51%, and children were 10%

Reviewer response: Please include this information in the manuscript.

Reviewer comment: The authors note that these are participants who were previously treated. Did all complete treatment or did some default on treatment? How many episodes have the previously had? If available, it would be worth including this information in the demographic section. Throughout this section it would be helpful to see the age of the participants along with the gender when quotes are provided.

No response given from authors.

Reviewer comment: The authors describe how side effects such as vomiting hinders adherence. However, it is not clear how sub-optimal dosing is a barrier to treatment adherence. While this is a clinical concern, it is a consequence rather than a barrier.

Author response: sub-optimal dose is a barrier to successful treatment completion.

Reviewer feedback: This is insufficient and is not demonstrated in either the data or the response.

Reviewer comment: Sub-theme 2.1. Consulting after a long illness It is not clear how this theme presents a barrier to adherence. Do the authors suggest that illness severity/progression is a barrier to adherence? It also seems similar to theme 2.2. The quotations illustrate that TB is hard on the body but not how it relates to treatment adherence.

Author response: Consulting after a long illness when the disease becomes serious, causing harm to their lungs, might hinder successful treatment completion and cure. It is not about adherence.

Reviewer response: Noted that it is about treatment completion and not adherence. However, the data still does not show HOW it hinders completion.

Reviewer #3: Congratulations to the co-authors on conducting a sound and important study on TB survivors' perspectives on barriers to successful treatment completion. This is important work that could be of policy / practice relevance.

However, I find the presentation of the manuscript at current as unacceptable for publication. This starts from the first two paragraphs in the abstract being duplicated, and continues into the reference list where the formatting has multiple font changes and errors. Overall, this still requires a careful copy-edit of entire text for clarity and precision. By way of example, in sub-section 2.2, we are twice told four hospitals (each named / listed) are where the data were collected. This could be shortened by writing something like: "The study settings were four DR-TB treatment-initiating hospitals in Ethiopia. Saint Peter TB Specialised and Alert hospitals are located in Addis Ababa, while Adama Referral and Bishoftu General Hospitals are located in Oromia Region in Adama and Bishoftu towns, respectively.". But then this is further compounded when the very next sentence (sub-section 2.3) reads "Previously-treated DR-TB patients at Bishoftu, Saint Peter, Alert and Adama hospitals, who were willing to provide informed consent to participate in the study and who were 18 years and above were included in the study." - why would we need the four hospitals listed a third time in three sentences? And then even this is compounded even further when information from that sentences (participants had to be >17-years-old) is repeated again by presenting the logical extrapolation that people "less than 18 years were excluded from the study". And as a final compounding of the problem, the paragraph ends with a sentence new to the revision of "Purposive sampling was used to recruit study participants.", but this not actually address the point raised by the reviewer that we need to know why these hospitals were selected, how patients treated at those hospitals were selected (i.e., for what purpose was the purposive sampling), nor the actual recruitment process (i.e., did you phone patients from information recorded on a register or something like that? If so, how many did not answer / declined participation etc.). Therefore, in these two sub-sections overall, we read a lot of repeated information that is not especially pertinent, and many of the details we actually need are not addressed. Again, this is an example of the overall need for copy-editing for precision. It is NOT limited to this one example. Frequently details already explained earlier in the text are repeated later.

In addition, I do not believe that this revision address the previous round of reviewer comments. Some comments seem to have been missed. In the response to reviewers, some of the authors' responses are in blue text, but others in black text, or missing altogether. Most responses just say "addressed", but do not indicate how, and it is often difficult to find track changes in the text that show how these are addressed.

I hope that the authors are encouraged by this feedback to do the work needed to revise the writing. As mentioned above, I believe that the study itself and the data merit publishing. I have found myself in the space of needing to do a complete edit on many papers that I've written. In the moment, this can feel overwhelming, and I have significant empathy for the authors. But in the overall scheme of the labour already exhausted on the project, it is only a small fraction to set aside a few more hours and push this over the line.

7. PLOS authors have the option to publish the peer review history of their article (what does this mean? ). If published, this will include your full peer review and any attached files.

**Do you want your identity to be public for this peer review?** For information about this choice, including consent withdrawal, please see our Privacy Policy .

Reviewer #2: No

Reviewer #3: **Yes: ** Graeme Hoddinott

While revising your submission, please upload your figure files to the Preflight Analysis and Conversion Engine (PACE) digital diagnostic tool, https://pacev2.apexcovantage.com/ . PACE helps ensure that figures meet PLOS requirements. To use PACE, you must first register as a user. Registration is free. Then, login and navigate to the UPLOAD tab, where you will find detailed instructions on how to use the tool. If you encounter any issues or have any questions when using PACE, please email PLOS at figures@plos.org. Please note that Supporting Information files do not need this step.

---

## [Decision Letter · Decision Letter 2]

6 Dec 2024

Treatment completion of drug-resistant tuberculosis in Ethiopia: a perspective from healthcare users

PGPH-D-23-02426R2

Dear Dr Tusho,

We are pleased to inform you that your manuscript 'Treatment completion of drug-resistant tuberculosis in Ethiopia: a perspective from healthcare users' has been provisionally accepted for publication in PLOS Global Public Health.

Best regards,

Leeberk Raja Inbaraj, MD

Academic Editor

Reviewer Comments (if any, and for reference):

Reviewer's Responses to Questions

**Comments to the Author**

1. If the authors have adequately addressed your comments raised in a previous round of review and you feel that this manuscript is now acceptable for publication, you may indicate that here to bypass the “Comments to the Author” section, enter your conflict of interest statement in the “Confidential to Editor” section, and submit your "Accept" recommendation.

Reviewer #3: All comments have been addressed

2. Does this manuscript meet PLOS Global Public Health’s publication criteria ? Is the manuscript technically sound, and do the data support the conclusions? The manuscript must describe methodologically and ethically rigorous research with conclusions that are appropriately drawn based on the data presented.

Reviewer #3: Yes

3. Has the statistical analysis been performed appropriately and rigorously?

Reviewer #3: N/A

4. Have the authors made all data underlying the findings in their manuscript fully available (please refer to the Data Availability Statement at the start of the manuscript PDF file)?

Reviewer #3: No

5. Is the manuscript presented in an intelligible fashion and written in standard English?

Reviewer #3: Yes

6. Review Comments to the Author

Reviewer #3: Well done for addressing the reviewers' comments.

The supplementary file is a statement on inclusivity in global health research. Where are the data uploaded? Given that they are qualitative data, should they be uploaded? Does that not risk breaking participant confidentiality?

The limitations section must be expanded to also consider the influence of group dynamics on what is reported in group discussions (as opposed to the more private individual in-depth interviews). E.g., Scheelbeek, P.F.D., Hamza, Y.A., Schellenberg, J. et al. Improving the use of focus group discussions in low income settings. BMC Med Res Methodol 20, 287 (2020). https://doi.org/10.1186/s12874-020-01168-8 . In this project, the findings are a long list of inoffensive possible barriers to treatment completion that the participants were happy to postulate. That is very different from a more critical / in-depth study to understand HOW each of these potential barriers have actually manifested in people's experiences, their relative importance, or synergies. Yet the discussion / conclusion accepts the list as 'the barriers', without further interrogation. I suggest that this needs further thinking through in the method, limitations, and that the conclusion be couched more as 'perceived potential barriers'.

7. PLOS authors have the option to publish the peer review history of their article (what does this mean? ). If published, this will include your full peer review and any attached files.

**Do you want your identity to be public for this peer review?** For information about this choice, including consent withdrawal, please see our Privacy Policy .

Reviewer #3: **Yes: ** Graeme Hoddinott
